# Learning Communication between Language Models through Dense Vectors

## Abstract

Communication between language models plays a crucial role in the inference process of large language models (LLMs), occurring both iteratively within a single model for multi-step reasoning (auto-regression) and interactively across multiple models for collaborative intelligence. While current systems primarily facilitate such communication through natural language, this paper proposes a novel paradigm of using continuous dense vector in continuous space. Our approach eliminates the unnecessary embedding and de-embedding steps when LLM interact with another, enabling more efficient information transfer, fully differentiable optimization path, and exploration of capabilities beyond human heuristics. We place such stripped LLMs as vertexes and optimizable seq2seq modules as edges to construct LMNet, a directed graph with similar structure as MLPs, and performs end-to-end gradient-descent for efficient optimization. As two exemplar applications, we show the proposed method can effectively improve LLM's general intelligence, and customizing LLM with limited data. We also provide detailed discussion and analysis about learning communication through dense vectors.

## 1 Introduction

Large Language Models (LLMs) have achieved impressive performance in natural language understanding, generation, and reasoning (Brown et al., 2020). Modern LLMs exhibit general intelligence capabilities across a wide range of subjects (Achiam et al., 2023; Yang et al., 2024; Grattafiori et al., 2024), but still face limitations when making inference for challenging tasks that require expertise knowledge or complex reasoning. The solutions for better inference can be summarized as designing communication protocol and structure among LLMs, which includes two perspectives. First considering the fact that auto-regressive generation can be viewed as a LLM iteratively communicating with itself, multi-step reasoning methods by single model such as chain-of-thought (Wei et al., 2022; Muennighoff et al., 2025; Zhang et al., 2025) are within this paradigm. Second, recent research has built multi-model/agent systems where multiple LLMs interact with each other and the environment (Hong et al., 2023;

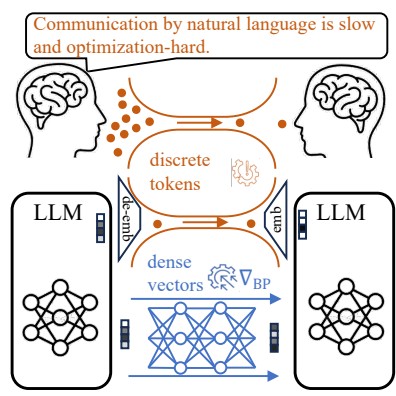

Figure 1: Communication between LLMs through dense vectors eliminates the bottleneck of natural language.

Zhou et al., 2025; Zhuge et al., 2024), which naturally rely on communication. And anything between them (Wang et al., 2022; Du et al., 2023) falls within this paradigm.

Most of existing systems use natural language as the medium of communication, due to the fact that LLMs are pre-trained on textual inputs and outputs. Natural language, however, is a symbolic and discrete representation, originally designed to align human understanding (Winograd, 1972; Chowdhary & Chowdhary, 2020). To be processed by LLMs, discrete tokens must be embedded into dense vectors before being passed through the model, and then decoded back into tokens at the output (Bengio et al., 2003; Mikolov et al., 2013; Vaswani et al., 2017). This introduces redundancy and inefficiency. As suggested by studies in cognitive science (Miller, 1956; Kahneman, 2011; Pinker, 2003; Donders, 1969), language is not an efficient way to transfer large amounts of information.

While natural language is essential for human interaction and model pre-training, it is not necessarily suitable for communication between LLMs.

This paper proposes a new paradigm of dense vector communication between LLMs. Instead of relying on discrete tokens, one LLM can output internal hidden states as dense vectors, and another LLM can directly accept these vectors as input—bypassing the embedding and de-embedding steps entirely. This mechanism offers several key advantages:

- **Higher information efficiency**: Dense vectors carry richer information per token, reducing loss from embedding and de-embedding.

- **Fully differentiable communication**: The system becomes end-to-end differentiable, supporting efficient gradient-based optimization.

- **Machine-native communication**: LLMs can learn roles and protocols suited for inter-model collaboration in more expressive hypothesis space, unconstrained by human-designed prior in natural language.

Under this paradigm, we are able to place LLMs as vertexes to construct and train a higher-level neural network, called LMNet. It can be formulated as a directed graph where each vertex is a stripped transformer without embedding or output layers, and each edge is a trainable sequence-to-sequence (seq2seq) module that transforms dense vectors between models. LMNet takes natural language as input and output, and is fully-differentiable. It can be applied for general NLP tasks. Two exemplar applications of the proposed method are studied empirically in this paper. The first one is improving LLM's general intelligence, i.e., training LMNet with large-scale pre-training datasets. It shows that LMNet can significantly improve a LLM's general reasoning ability, with small additional training cost which is less than 0.2% of pre-training the LLM. The second one is customizing LLM with limited data, where LMNet shows strong performance in scenarios including improving reasoning ability comparing with communication in natural language and latent reasoning methods, and serving the role comparable to parameter-efficient fine-tunings (PEFT) method to adapt pre-trained LLM with strictly-constrained data.

## 2 RELATED WORKS

We review existing works that construct communication mechanisms between LLMs for better inference. This involves designing both the topology and functional roles of inter-model communication. These efforts can be broadly categorized into two types: multi-step reasoning and multi-model collaboration.

Multi-step reasoning (Wei et al., 2022; Yao et al., 2023a; Besta et al., 2024; Tian et al., 2024), or more generally test-time scaling (Muennighoff et al., 2025), aims to improve LLM performance by allocating additional computational resources during inference. Exemplar approaches include: Chain-of-Thought (CoT) Prompting (Wang et al., 2022): Encourages LLMs to generate intermediate reasoning steps, improving accuracy on complex tasks at the cost of increased inference time; Parallel Sampling: like Majority Voting (self-consistency) (Wang et al., 2022; Suzuoki & Hatano, 2024) and Best-of-N Sampling (Wang et al., 2024b; Gui et al., 2024), which generates multiple outputs at a single step and selects the best by certain strategies, though this can be computationally expensive; Process-Based Verifiers (Setlur et al., 2024; Zhang et al., 2024a): which makes LLM generate intermediate reasoning steps with tree structure, and train models to evaluate intermediate steps (e.g., Process Reward Models), enabling more efficient tree-search strategies.

Multi-model collaboration (Talebirad & Nadiri, 2023; Liu et al., 2025; 2024; Zhang et al., 2024b) builds complex workflows, including multi-agent systems powered by LLMs, to improve performance by leveraging specialized LLMs to divide complex tasks into subtasks. They can either be built by expert knowledge from human's prior only, or data-driven. Without training data, exemplar works integrate standardized operating procedures into multi-agent workflows for software development (Hong et al., 2023), or design prompts to encourage LLM reasoning and acting with given tools iteratively (Yao et al., 2023b). Given training data from the specific domain for data-driven customization, existing works optimize by search (Khattab et al., 2024; Zhuge et al., 2024; Zhang et al., 2024c; Zhou et al., 2025) or using LLM as black-box optimizer (Liu et al., 2024).

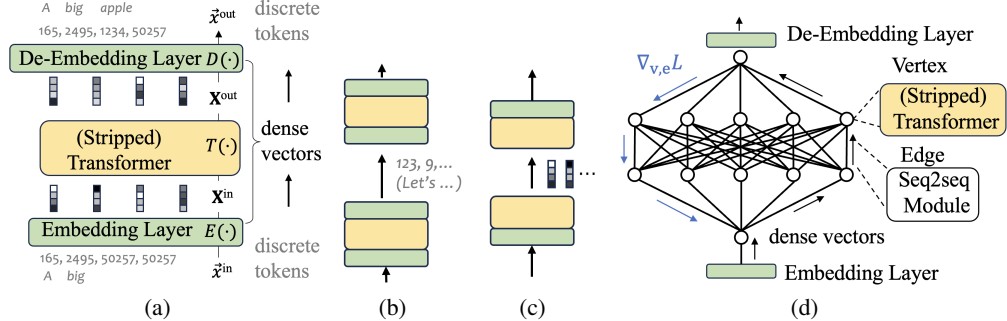

Figure 2: Illustration of the proposed paradigm. (a) A standard LLM processes discrete token inputs by embedding them into dense vectors, and outputs discrete tokens via a de-embedding layer. (b) Existing communication between LLMs typically occurs through discrete tokens. (c) Our approach strips the embedding and de-embedding layers, allowing LLMs to communicate directly via dense vectors. (d) We construct and train a LMNet by connecting stripped transformers with trainable communication modules.

Despite their diversity, all of the above methods rely on natural language communication using discrete token sequences. This introduces inefficiencies in both information transfer and optimization. Two groups of works partially move beyond this limitation. There are works focusing on chain-of-thought reasoning in latent space rather than natural language (Hao et al., 2024; Cheng & Van Durme, 2024; Shen et al., 2025), but they are limited to recurrent inference within a single LLM and cannot generalize to more complex reasoning topologies, heterogeneous model interactions, or multi-agent collaboration. They will be specially compared in comparable setting (Section 4.2). Another line of work explores differentiable communication among reinforcement learning agents via a centralized neural controller, but this approach is not applicable to pre-trained LLMs or natural language tasks (Sukhbaatar et al., 2016).

## 3 METHOD

To enable the advantage of learning machine-native communication by laying LLMs in an expressive enough hypothesis space of communication, unconstrained by human-designed prior in natural language, we place LLMs as vertexes to construct a higher-level neural network, LMNet.

### 3.1 CONSTRUCTING LMNET

The construction includes (i) stripping de-embedding layers from LLMs to serve as differentiable vertexes; (ii) introducing edge modules to make connection; and (iii) defining communication topology. Details on the specification of the modules (vertex, edge, and aggregation) as well as the topology hypothesis are provided in the Appendix A.

#### 3.1.1 VERTEX: STRIPPING EMBEDDING LAYERS FROM TRANSFORMERS

Denote a tokenized text in natural language as $\vec{x}^{\text{in}} = [x_1, x_2, \cdots, x_n]$, a sequence of discrete tokens where each token $x_i \in \mathcal{D}$ and $|\mathcal{D}|$ is the vocabulary size. The embedding layer $E$ embeds the discrete tokens into dense vectors, $\mathbf{X}^{\text{in}} = E(\vec{x}^{\text{in}})$, where $\mathbf{X}^{\text{in}} = [\mathbf{x}_1, \mathbf{x}_2, \cdots, \mathbf{x}_n]$ and $\mathbf{x}_i \in \mathbb{R}^d, d << |\mathcal{D}|$. Inside a LLM, the transformer model $T$ takes $\mathbf{X}^{\text{in}}$ as input, and output dense vectors with the same size, denoted as $\mathbf{X}^{\text{out}} = T(\mathbf{X}^{\text{in}})$. The de-embedding layer $D$ decodes the dense vectors to discrete tokens to output natural language text, denoted as $\vec{x}^{\text{out}} = D(\mathbf{X}^{\text{out}})$. The complete function of a LLM $f$ is $\vec{x}^{\text{out}} = D \circ T \circ E(\vec{x}^{\text{in}}) = f(\vec{x}^{\text{in}})$.

Given two LLMs $f_1, f_2$ with communication flow $f_1 \rightarrow f_2$, the existing and natural way is to let them communicate with natural language by setting $\vec{x}_2^{\text{in}} = \vec{x}_1^{\text{out}}$, i.e., $\vec{x}^{\text{out}} = D_2 \circ T_2 \circ E_2 \circ D_1 \circ T_1 \circ E_1(\vec{x}^{\text{in}}) = f_2 \circ f_1(\vec{x}^{\text{in}})$. Note that rather than merely a feeding-forward module, $D$ includes discretization operation like $\arg\max$, which leads to undesired information loss and cutoff of gradient. Therefore, we propose to remove the internal de-embedding layer $D_1$ and correspondingly removing embedding layer $E_2$. We let them communicate with dense vectors by setting $\mathbf{X}_2^{\text{in}} = \mathbf{X}_1^{\text{out}}$, i.e., $\vec{x}^{\text{out}} = D_2 \circ T_2 \circ T_1 \circ E_1(\vec{x}^{\text{in}})$. More generally, for communication flow $f_1 \rightarrow f_2 \cdots \rightarrow f_n$, all internal de-embedding layer and embedding layers will be removed, i.e., $\vec{x}^{\text{out}} = D_n \circ T_n \circ T_{n-1} \circ \cdots \circ T_1 \circ E_1(\vec{x}^{\text{in}})$: there are only one embedding layer $E_1$ and one de-embedding layer $D_n$ to keep natural language

input-output of the complete system, while all intermediate LLMs are stripped transformer $T$, communicated through dense vectors. This eliminates internal information loss and enables joint gradient descent for efficient optimization.

### 3.1.2 EDGE: TRAINABLE SEQ2SEQ MODULES FOR CONNECTION

Another critical component in our method is communication module, or edge module $e$. Each $e$ is a small seq2seq module which will function on a communication path $\mathbf{X}_i^{\text{in}} = e(\mathbf{X}_{i-1}^{\text{out}})$. Note that the stripped transformer $T_i$ from a pre-trained LLM has not learned to take dense vectors $\mathbf{X}_i^{\text{in}} = \mathbf{X}_{i-1}^{\text{out}} \in \mathbb{R}^d$ as input yet, as it is pre-trained with input from a discrete space $\mathbf{X}_i^{\text{in}} = E_i(\vec{x})$ which is very different. Introducing $e$ is not only aimed at aligning them, but also enabling translating and distributing a single $\mathbf{X}_{i-1}^{\text{out}}$ to multiple targets differently with multiple different $e$. This is the key to enable light-weight learning dense communication, and complex communication topology.

### 3.1.3 COMMUNICATION TOPOLOGY

While it is typical to formulate the hypothesis of workflow of LLM systems as a directed graph, where vertexes are LLMs and edges are communication paths, we specify it as a layer-wise fully connected feed-forward network for simplicity. This is inspired by the similar topology among neurons in MLP, which enables the universal approximation property (Hornik et al., 1989). Formally, denote LMNet as $\mathcal{N} = (\mathcal{V}, \mathcal{E})$ where:

- Each vertex $v \in \mathcal{V}$ represents $T$ (a stripped transformer without embedding and de-embedding layer). The vertexes are arranged in $L$ layers: $\mathcal{V} = \{\mathcal{V}^l\}_{l=1}^L = \{\{v_i^l\}_{i=1}^{n_l}\}_{l=1}^L$. Specially, there is only one vertex at the last layer as the output vertex, $n_L = 1$, whose de-embedding layer $D_1^L$ is kept to convert dense vectors to discrete tokens in natural language for final output. And there is only one vertex at the first layer as the input vertex, $n_1 = 1$, whose embedding layer $E_1^1$ are kept to convert discrete tokens in initial input natural language to dense vectors. We denote $v_1^0 = E_1^1$ for notation consistency.

- Each edge $e \in \mathcal{E}$ represents a communication path, through a trainable seq2seq module parameterized by $\omega$. As a layer-wise fully connected structure, there are and only are $\{e_{i,j}^l \mid \forall v_i^l \in \mathcal{V}^l, \forall v_j^{l+1} \in \mathcal{V}^{l+1}\}$.

Given initial input text $\vec{x}^{\text{in}}$, the embedding layer $E_1^1$ embeds it to dense vectors $\mathbf{X}^{\text{in}}$. Then, $\mathcal{N}$ works in the way like a feed-forward neural network does. Formally, initializing $\mathbf{X}_1^0 = \mathbf{X}^{\text{in}}$, we have

$$\mathbf{X}_j^{l+1} = v_j^{l+1}\left(\sum_{v_i^l \in \mathcal{V}^l} e_{ij}^l(\mathbf{X}_i^l; \omega_{ij}^l); \theta_j^{l+1}\right), \tag{1}$$

where $\theta_j^{l+1}$ is the parameter in vertex module $v_j^{l+1}$ and $\omega_{ij}^l$ is the parameter in edge module $e_{ij}^l$. The final output $\mathbf{X}_1^L$ is de-embedded to text $\vec{x}^{\text{out}} = D_1^L(\mathbf{X}_1^L)$, and mapped to the output logits $\boldsymbol{p}^o \in \mathbb{R}^{n \times |\mathcal{D}|}$ for optimization. In this way, such a LMNet takes natural language as input and output in the same way as a LLM system typically does, thus can be applied for general NLP tasks.

### 3.2 TRAINING

One of the most significant benefits of communication through dense vectors is that the path is differentiable. Given a differentiable supervision signal $\mathcal{L}$, i.e., $\partial \mathcal{L}/\partial \boldsymbol{p}^o$, we can obtain gradient on all parameters in LMNet, i.e., $\partial \mathcal{L}/\partial \theta_i^l$ and $\partial \mathcal{L}/\partial \omega_{ij}^l$ for any $l, i, j$. This enables joint and efficient optimization by end-to-end gradient descent. Thus, all stages in the pipeline of training a single LLM can be applied to train LMNet, including auto-regressive pre-training, supervised fine-tuning, training by reinforcement learning. The training data and strategies are to be determined specifically to the application. We do not specify here but will implement exemplar applications in Section 4 by performing end-to-end auto-regressive training.

It is recommended to treat $\boldsymbol{\theta}$ and $\boldsymbol{\omega}$ differently, as $\boldsymbol{\theta}$ has already been pre-trained with fine-grained information, while $\boldsymbol{\omega}$ is randomly initialized that contains no basic knowledge at all. It is recommended to first do pre-training with naive auto-regressive loss optimizing $\boldsymbol{\omega}$ only, to equip LMNet with basic ability to model natural language. The inference strategy is coupled with the training.

In this paper we would only implement end-to-end auto-regressive training and decoding for inference. It is also expected to be powerful to make inference by letting each vertex decode to generate complete/multiple-token sequence, then feeding to latter modules. This can be achieved by implementing such process while still supervise the final output only through post-training.

## 3.3 Reducing Complexity by Vertex Parameter Sharing

It is undeniable that LMNet increases complexity comparing with a single vertex LLM, both in parameter size and inference latency per token. With the proposed parameter-sharing technique, the cost in both aspects may be far less than anticipated. Denote the parameters in LMNet as $\boldsymbol{\theta}$ and $\boldsymbol{\omega}$ as the collection of vertex and edge parameters respectively. Denote the depth of LMNet as $L$ and average width as $W$, approximately with $L \times W^2$ edges following the layer-wise fully-connected topology. Denote the average time-consumption of feeding-forward as $t_\theta$ through a vertex and $t_\omega$ through an edge. With vertex parameter sharing, which means given a single pre-trained LLM $\theta_v$, i.e., initializing all $\theta_i^l$ with $\theta_v$ and keeping $\theta_i^{l_1} = \theta_j^{l_2}$ for any $l_1, l_2, i, j$ during training, the parameter size is $|\boldsymbol{\theta}| + |\boldsymbol{\omega}| = |\theta_v| + L \times W^2 \times |\omega|$ where $|\omega| << |\theta_v|$. This means when the size of LMNet grows, its parameter size only grows by a small ratio. This gives the possibility to build a deep and wide LMNet. For time complexity, keeping vertexes in the same layer identical enables simply paralleling them through 'torch.DataParallel'. In this case, a feeding-forward process only requires a sequential processing with length $L$, like a MLP, and $t = L \times t_\theta + L \times W^2 \times t_\omega$, where $t_\omega \ll t_\theta$.

## 4 Experiments

In this section, we discuss two exemplar applications of LMNet and provide empirical results. First, we show the effect of **learning dense communication for general intelligence**. We do our best in a data-rich way to build a LMNet that can understand human's instructions for diverse tasks and generate coherent natural language. Then, we show the application of **customizing LLM with limited data**. Given limited training set on a domain-specific problem, we build LMNet to adapt to the given training data. Code is provided at `https://anonymous.4open.science/r/LMNet_Code-231E`.

### 4.1 Learning Dense Communication for General Intelligence

Society of Mind theory (Minsky, 1986) suggests that higher-level intelligence emerges from the coordination of simpler components. LMNet can embody such idea of collective intelligence, by coordinating pre-trained LLMs: each vertex (a stripped transformer from pre-trained LLM) acts as a modular component, while the edges (trainable seq2seq modules) facilitate differentiable communication, enabling the system to collectively solve problems beyond the reach of any single LLM. By optimizing these communications, we hope that LMNet not only scales computational power but also fosters emergent behaviors, such as advanced reasoning and adaptive problem-solving, that transcend the capabilities of its individual components.

#### 4.1.1 Settings

Due to the computation budget, we consider modern LLMs with minimum size. We use Qwen2.5-0.5B as vertex module to be shared among all vertexes. We implement LMNet with 5 layers with 1/4/4/4/1 vertexes in each layer. We use an attention block with the same structure as one transformer layer in the vertex module for each edge module. These edge modules are independently random initialized and to be optimized. Such a LMNet has 1.1B parameters in total. All data used in this study come from public datasets without testing leakage. The details are provided in Appendix B. As mentioned above, we first freeze vertex parameters and only update edge parameters with the typical auto-regressive loss, then we update all parameters together.

We compare the trained LMNet with other solutions based on Qwen2.5-0.5B, including **Prompt** (using ad-hoc prompt for best performance), and **SFT** (using the same training data as LMNet to fine-tune all parameters of a Qwen2.5-0.5B). We evaluate the trained LMNet on widely-used benchmarks, with details provided in Appendix B. Due to computation budget, we report the performance of LMNet trained for less than 100 GPU·days (NVIDIA A100), and less than 0.02T tokens or 2e5

Table 1: Performance comparison for improving general intelligence with LMNet. (Accuracy, %).

| Model | Qwen2.5-0.5B | | | Llama3.2-1B | Qwen2.5-1.5B | Gemma2-2B | Llama3.2-3B |
|---|---|---|---|---|---|---|---|
| Method | Prompt | SFT | **LMNet** | Prompt | Prompt | Prompt | Prompt |
| # Parameters | 0.49B | 0.49B | 1.14B | 1.23B | 1.54B | 2.61B | 3.21B |
| **General Tasks** — MMLU | 44.3 | 44.6 | 53.9 | 32.2 | **60.9** | 52.2 | 58.0 |
| MMLU-pro | 15.7 | 13.2 | 26.2 | 12.0 | **28.5** | 23.0 | 22.2 |
| BBH | 20.3 | 19.9 | **47.3** | 31.6 | 45.1 | 41.9 | 46.8 |
| ARC-C | 35.6 | 34.6 | 38.0 | 32.8 | 54.7 | 55.7 | **69.1** |
| Truthfulqa | 40.2 | 40.5 | **47.9** | 37.7 | 46.6 | 36.2 | 39.3 |
| **Math & Science** — GSM8K | 41.6 | 41.8 | 50.3 | 9.2 | **68.5** | 30.3 | 12.6 |
| MATH | 19.5 | 12.4 | **38.8** | - | 35.0 | 18.3 | - |
| GPQA | 24.8 | 22.0 | **25.6** | 7.6 | 24.2 | 25.3 | 6.9 |
| MMLU-stem | 39.8 | 39.8 | 46.0 | 28.5 | **54.8** | 45.8 | 47.7 |
| **Coding** — HumanEval | 30.5 | 27.6 | **39.0** | - | 37.2 | 19.5 | - |
| MBPP | 39.3 | 35.1 | 45.8 | - | **60.2** | 42.1 | - |
| 40.1% relative performance gain, 0.2% additional training cost | | | | | | | |
| | | | Better/comparable performance with models in similar/larger size | | | | |

PFLOPs in total. In terms of either tokens or computation cost or time, the training cost of LMNet is significantly less (< 0.2%) than the pre-training cost of LLMs.

### 4.1.2 PERFORMANCE

The performance results are provided in the left half of the Table 1. First, LMNet brings consistent and significant improvement comparing with Prompt, which can be viewed as communication through natural language (recurrently with itself). This indicates the advantage of proposed communication medium and topology, and gives a way to improve general performance given a pre-trained LLM, using public data and acceptable training cost. Second, SFT(implemented by very moderate training hyperparameters) can hardly bring improvement over the pre-trained LLM. In fact, these public data is likely to be seen during the pre-training of LLM. This indicates the benefit of LMNet comes from the learned communication, rather than the training data.

Considering the fact that LMNet increases parameter size to 1.1B while the other two baselines are 0.5B, we also compare the trained LMNet with modern pre-trained LLMs in similar size, including Qwen-1.5B (Yang et al., 2024), Llama3.2-1B/3B (Grattafiori et al., 2024), Gemma2-2B (Team et al., 2024),. The results are provided in Table 1 right half, where LMNet shows comparable or even better performance. Note that this paper aims to provide a method rather than the trained LMNet ready-to-use. In Appendix B.2, we discuss the opportunities of LMNet as a method for scaling for general intelligence, which show advantage in training cost and data consumption comparing with training a monolithic LLM from scratch.

### 4.1.3 ANALYSIS OF THE LEARNED DENSE COMMUNICATION

We take a closer look at the trained LMNet, to see what communication has been learned that improves LLM's general intelligence de facto.

**On Macro-Level**   First, we try to find some topology patterns across the edges. by visualization and analysis the parameters in all edge modules of the edges connecting the 1/4/4/4/1 vertexes. We reach a conclusion that the layer-wise fully connected structure is fully exploited through training, without distinguishable substructures. Detailed results are provided in Appendix C.1.

**On Micro-Level**   Second, we perform case study to see the inference process. The case is the first test case in GSM8K, with input text shown in blue block in Figure 3. The trained LMNet answers correctly with output text shown in the green block.

To see if the communication makes effect, we first try to de-embed the input dense vector sequences of different intermediate vertexes, i.e., the output vectors of edge modules. But we failed with most tokens typically shows very close logits on many words, and even if we force to find a word with

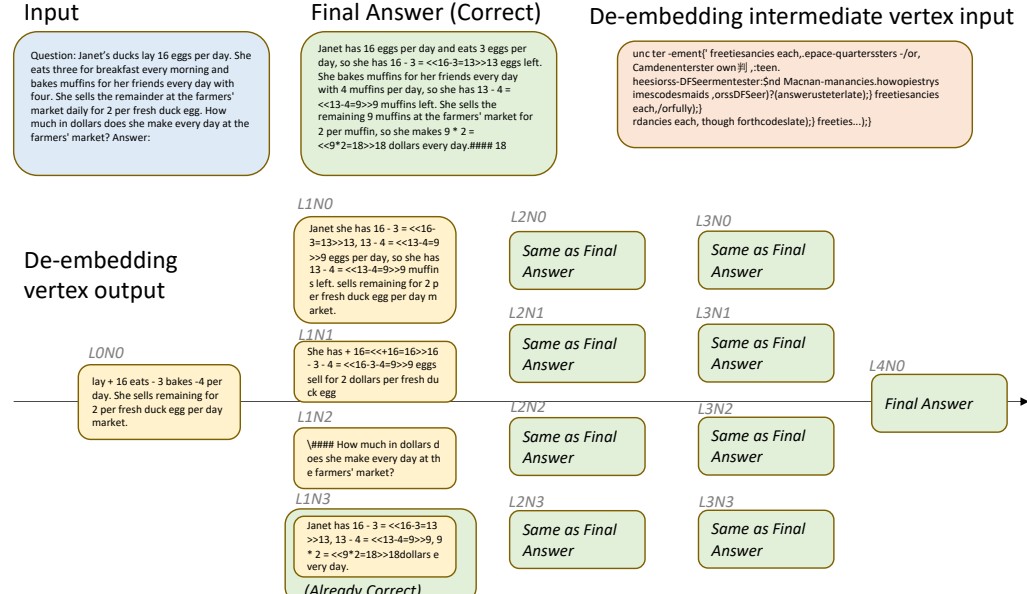

Figure 3: Case study of learned communication in trained LMNet.

hard-max logit, the outcome input text is not even English (e.g., the red block in Figure 3). However, we find the output vectors of intermediate vertexes, i.e., the input vectors of edge modules, can be de-embedded to meaningful sentences. As shown in the lower part of Figure 3, we de-embed the output sentences of all the 1/4/4/4/1 vertexes and truncate to visualize. Details about how to de-embed the intermediate dense vectors and process are provided in Appendix C.2. We have the following observation and conclusions:

- The trained LMNet makes inference generally in the expected "division/multi-step reasoning" way through communication. Correct and confident answer can be achieved by early vertexes for easy problem. In this case, the correct answer "18" is first output by L1N3, and all the later vertexes outputs the same sentence as the final output.

- Intermediate vertexes, especially earlier vertexes, no longer necessarily output complete answer or coherent sentence. They tend to aggregate important information at very early positions in the sentence, thus output shorter, comparing with the output vertex. This is because of the end-to-end auto-regressive manner. Only in this way the final output is generated with taking the important information output by the intermediate vertex

- The input space of the intermediate vertexes has departure far from the word embeddings, but the output space of them remains similar. The input space departure can be interpreted by that we implement the topology with layer-wise fully connected and the aggregation function of different input edges on a single vertex with simply summation, that fuses a lot; and the joint optimization of vertex, edges, embedding parameters. The output space are similar can be interpreted as the parameter-sharing of vertex modules: the LLM as intermediate vertexes are the same one as the final output vertex, which are supervised to generate natural language by the word embeddings.

## 4.2 CUSTOMIZING LLM WITH LIMITED DATA

Now we discuss another scenario where LMNet can be effectively applied: adapting pre-trained LLM with limited data. We consider fair comparison under a strict setting that only a pre-trained LLM and a training set are accessible, to improve the performance on unseen testing set from the same domain. Under this setting, the common practice is parameter-efficient fine-tuning (Houlsby et al., 2019; Li & Liang, 2021; Hu et al., 2022), to avoid over-fitting the very limited data. The importance of collective intelligence (Minsky, 1986) and structure of LMNet gives an option: we can build a collective intelligence system and let the limited data determines the communications, by constructing a LMNet with shared vertex parameters as regularization, and/or freezing the vertexes with weights from pre-trained LLMs, training few edge parameters only.

Table 2: Performance comparison on MMLU dataset ($\Delta$Acc, %).

| | GPT2-XL | Llama3.2-1B | Llama3.2-1B-Instruct | Qwen2.5-0.5B | Qwen2.5-1.5B |
|---|---|---|---|---|---|
| Pred | 0.24 | 6.05 | 20.93 | 22.36 | 34.75 |
| Prompt | 1.69 | 11.69 | 24.30 | 22.50 | 35.83 |
| SFT | 7.40 | 24.65 | 24.83 | 21.50 | 35.02 |
| COCONUT | 3.63 | 9.88 | 18.54 | 19.22 | 33.85 |
| CODI | 5.01 | 20.69 | 21.30 | 19.48 | 35.16 |
| **LMNet** | **13.10** | **27.19** | **27.57** | **23.35** | **37.28** |

Table 3: Performance comparison on GSM8K dataset (Accuracy, %).

| | Llama3.2-1B | Llama3.2-1B-Instruct | Qwen2.5-0.5B | Qwen2.5-1.5B |
|---|---|---|---|---|
| Pred | 2.88 | 30.10 | 5.31 | 9.25 |
| Prompt | 11.49 | 43.67 | 41.60 | 68.50 |
| SFT (w/ CoT) | 38.69 | 46.39 | 42.94 | 69.31 |
| COCONUT | 26.79 | 38.27 | 30.81 | 48.63 |
| CODI | 37.25 | 44.58 | 42.06 | 65.30 |
| **LMNet** | **44.02** | **56.15** | **50.82** | **72.67** |

We consider fair comparison under a strict setting that only a pre-trained LLM and a training set are accessible, to improve the performance on unseen testing set from the same domain. Generally, we can consider three categories of methods: (i) prompting, including demonstrating examples from training set for ICL and prompting to reasoning better like CoT ; (ii) fine-tuning, including learning to latent reasoning, and parameter-efficient fine-tuning methods; (iii) training LMNet. We consider benchmark datasets MMLU, GSM8K and E2E (Novikova et al., 2017) respectively.

### 4.2.1 IMPROVING REASONING ABILITY ON MMLU AND GSM8K

We study on widely used benchmark MMLU and GSM8K to show the effect of LMNet on different models. The baselines include **LMNet**: construct LMNet with the given LLM as vertex and train with training set. **Pred**: directly ask the LLM to answer the question; **Prompt**: use model- and dataset- specific prompt engineering to optimize the performance (typically and at least combines ICL and CoT); **SFT**: fine-tune the LLM with the training set, including learning for latent reasoning. Specifically, as these two datasets focus on the reasoning ability, there are two additional baselines which do latent reasoning, which can be viewed as learning communication in latent space rather than natural language: **COCONUT** (Hao et al., 2024) and **CODI**(Shen et al., 2025). The difference between LMNet between them is LMNet enabling complex topology and trained end-to-end auto-repressively, while they only communicate iteratively with the LLM itself, trained by un-supervising some tokens or self-distillation.

For both datasets, we construct LMNet with a small scale (3 layers with 1/2/1 vertexes), and keep all the vertexes share the same group of parameters for efficiency. All parameters in LMNet are updated together by gradient descent. We use a module with the same structure with a single transformer layer from the corresponding backbone LLM. Taking Qwen2.5-1.5B for example, a single LLM model has $1.76 \times 10^9$ parameters, while constructing the LMNet only introduce additional $2.45 \times 10^8$ parameters on the edge modules. In this case, the parameter size of LMNet is close to a single model, and the cost of training LMNet and fine-tuning a single model have similar scale. The performance on MMLU is evaluated by $\Delta$Acc, which is the difference between testing accuracy and random guess average accuracy (25%), provided in Table 2. For GSM8K, we noticed that the training set of GSM8K is much smaller than MMLU (7.47k vs 100k), which seriously constrains the methods' effect, as reported in Appendix D. So we add additional 93.7k training data from OpenR1-Math-220k[1] for all baselines requiring training. Results are provided in Table 6.

LMNet shows general and relatively significant performance advantage over the other baselines with every considered LLM on both benchmarks. Specially comparing LMNet with COCONUT and CODI, as they can also be viewed as in the paradigm communication through dense vectors, they can only express chain (-of-thought) reasoning structure, and are trained to mimic the process of explicit CoT in natural language, resulting in no better than the performance of CoT. On the contrary, LMNet are equipped with more complex reasoning structure by defining the fully-connected topology hypothesis, and are trained without supervision on the intermediate communication process, contributing to the advantage in performance.

---

[1] https://huggingface.co/datasets/open-r1/OpenR1-Math-220k

Table 4: Performance comparison on E2E dataset with GPT2-M. * indicates results from (Hu et al., 2022).

| Method | Metrics ↑ | | | | | Rank Avg. |
|---|---|---|---|---|---|---|
| | BLEU | NIST | MET | ROUGE-L | CIDEr | |
| No Adaptation | 0.00 | 0.42 | 0.04 | 0.16 | 0.00 | 9.0 |
| FT* | 68.2 | 8.62 | 46.2 | 71.0 | 2.47 | 4.5 |
| Adapter$^L$(0.37M)* (Lin et al., 2020) | 66.3 | 8.41 | 45.0 | 69.8 | 2.40 | 8.0 |
| Adapter$^L$(11.09M)* (Lin et al., 2020) | 68.9 | 8.71 | 46.1 | 71.3 | 2.47 | 3.9 |
| Adapter$^H$* (Houlsby et al., 2019) | 67.3 | 8.50 | 46.0 | 70.7 | 2.44 | 6.7 |
| FT$^{Top2}$* (Li & Liang, 2021) | 68.1 | 8.59 | 46.0 | 70.8 | 2.41 | 6.3 |
| PreLayer* (Li & Liang, 2021) | 69.7 | 8.81 | 46.1 | 71.4 | 2.49 | 2.7 |
| LoRA (Hu et al., 2022) | 68.9 | 8.68 | **46.5** | **71.5** | **2.51** | 2.3 |
| **LMNet** | **70.5** | **8.85** | **46.5** | **71.5** | 2.48 | **1.6** |

### 4.2.2 DATA-EFFICIENT ADAPTATION ON E2E DATASET

A common issue in adapting/fine-tuning LLM is the data scarcity, comparing with the massive parameter amount. Such issue is typically addressed through PEFT methods, which identify few core parameters to make effective adaptation while avoid over-fitting. LMNet gives another option that learns the data-dependent communication. In this part, we make comparison with PEFT methods following the experiment setting in LoRA(Hu et al., 2022): we study with GPT2-M (Radford et al., 2019) model on E2E dataset, which is known to require effective adaptation with limited data, and consider the same group of PEFT methods (Houlsby et al., 2019; Li & Liang, 2021; Hu et al., 2022).

To avoid over-fitting with LMNet, by default we only train the edge parameters and keep the vertex parameters frozen. We still construct LMNet with a small scale (3 layers with 1/2/1 vertexes), using an attention block, containing 5.25M parameters, for each edge module. The results are provided in Table 4. LMNet performs best, showing learning dense communication makes effective and generalizable adaptation. Thanks to the fully-differentiable paths in LMNet, we can also integrate LMNet along with PEFT methods. Results in Appendix E suggest that plugging-in appropriate PEFT methods can further improve the performance.

## 5 CONCLUSION, LIMITATIONS AND FUTURE WORKS

In this paper, we introduce LMNet, a novel paradigm building communication between LLMs through dense vectors, by stripping the embedding and de-embedding layers and place LLMs as vertex in a directed graph with similar structure as a MLP. LMNet constructs collective intelligence system which facilitates more efficient information transfer, fully differentiable optimization path, and the exploration of capabilities beyond human heuristics. As exemplar applications, we show the proposed method can effectively improves LLM's general intelligence, and customize LLM with limited data.

The most noteworthy limitation of the LMNet is its complexity. As discussed in Section 3.3, it unavoidably increases parameter size and inference latency. Though LMNet enlarges parameter size comparing its vertex LLM, it shows performance advantage over not only its vertex LLM, but also other monolithic LLM with similar size. The inference latency per token could be about LMNet layer number times vertex latency, with certain implementation of parameter-sharing. But considering the performance difference and many factors contributing to the overall inference latency (e.g., output length, parameter size, model structure), better performance requires more latency in general. We may explore more wide but less deep architecture of LMNet, and training with less output length in the future, to reduce inference latency.

The current paper only provides the basic idea and primary practice. Due to our budget on computation resources, we could not have ablation study to refine the design and verify the generalizability of effect on other pre-trained LLMs, but it can be expected that the principle of learning dense communication does generalize. There are many directions for further exploration, including specification of modules, topology hypothesis and training/decoding process. LMNet can also be equipped with multiple input/output vertexes to interact with environment to build LLM-based agent systems. (Post-)Training and inference (decoding) strategies beyond end-to-end auto-regressive generation are also left to explore. We leave exploring larger-scale collection of versatile vertexes, training with refined training protocols, to release a well-trained strong LMNet ready-to-use, to future works.

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

# A  DISCUSSION

**Specification of Modules**   For edge modules, any seq2seq modules are applicable. Note that this can be reduced to element-wise functions like a MLP processing each vector independently. To be parameter-efficient yet expressive enough, we choose to use a single attention-block (1-layer transformer) for each edge. All edge modules would be independently and randomly initialized. For vertex models, LMNet does not require all the vertex models to be identical. We can implement each vertex with different pre-trained LLM respectively, to exploit that different LLMs may have different expertise and LMNet can combine them together. However, we can choose a much more parameter-efficient way: implementing all the vertex with a single pre-trained LLM. As different edges are different, the input information of different vertex would be different, so LMNet would still be effective. For the aggregation function of messages from multiple input edges of one vertex, we use sum for simplicity: in this case we can use the same causal mask for all vertexes and edges, to keep the causal structure of the input sequence of pre-trained LLMs.

**Topology Hypothesis**   We construct the layer-wise fully connected structure as the hypothesis of workflow of a LLM system by default. This is inspired by the similar topology among neurons in MLP. Akin to the universal approximation property of MLPs, LMNet can express a wide range topology and functions of workflows. For example, one can easily figure out chain/tree/graph structures inside the fully-connected structure. There are also many other reasonable topology choices, like several paths with no intermediate intersection, skip or residual connection, or mimicking message passing on given graph. One can design the topology according to problem-specific prior knowledge, expertise in vertexes, and expressiveness theory.

# B  COMPARING LMNET WITH MONOLITHIC LLMS IN SIMILAR SIZE

All data used in this study come from public datasets: C4 (Raffel et al., 2020), Alpaca (Taori et al., 2023), ProsocialDialog (Kim et al., 2022), LaMini-instruction (Wu et al., 2024), MMLU (Hendrycks et al., 2021a) (auxiliary_training split only), MATH (Hendrycks et al., 2021b) (training split only), GSM8K (Cobbe et al., 2021) (training split only). Note that due to computation budget, we report the performance of LMNet trained for less than 100 GPU·days (NVIDIA A100), and less than 0.02T tokens or 2e5 PFLOPs in total. This means only a small subset of above mentioned datasets are used.

## B.1  PERFORMANCE OF (RE)PRE-TRAINED LMNET

Table 5: Performance comparison of pre-trained LLMs (Accuracy, %).

| Model | | Qwen2.5-0.5B | **LMNet-1B** | Llama3.2-1B | Qwen2.5-1.5B | Gemma2-2B | Llama3.2-3B |
|---|---|---|---|---|---|---|---|
| # Parameters | | 0.49B | 1.14B | 1.23B | 1.54B | 2.61B | 3.21B |
| # Training Tokens | | 18T | +0.02T | 15T | 18T | 15T | 15T |
| General Tasks | MMLU | 44.3 | 53.9 | 32.2 | **60.9** | 52.2 | 58.0 |
| | MMLU-pro | 15.7 | 26.2 | 12.0 | **28.5** | 23.0 | 22.2 |
| | BBH | 20.3 | **47.3** | 31.6 | 45.1 | 41.9 | 46.8 |
| | ARC-C | 35.6 | 38.0 | 32.8 | 54.7 | 55.7 | **69.1** |
| | Truthfulqa | 40.2 | **47.9** | 37.7 | 46.6 | 36.2 | 39.3 |
| Math & Science | GSM8K | 41.6 | 50.3 | 9.2 | **68.5** | 30.3 | 12.6 |
| | MATH | 19.5 | **38.8** | - | 35.0 | 18.3 | - |
| | GPQA | 24.8 | **25.6** | 7.6 | 24.2 | 25.3 | 6.9 |
| | MMLU-stem | 39.8 | 46.0 | 28.5 | **54.8** | 45.8 | 47.7 |
| Coding | HumanEval | 30.5 | **39.0** | - | 37.2 | 19.5 | - |
| | MBPP | 39.3 | 45.8 | - | **60.2** | 42.1 | - |

We compare the trained LMNet in Section 4.1 (shared Qwen2.5-0.5B as vertexes, 1/4/4/4/1 structure. 1.1B parameters in total) with modern pre-trained-only LLMs in similar size, including Qwen-0.5B/1.5B (Yang et al., 2024), Llama3.2-1B/3B (Grattafiori et al., 2024), Gemma2-2B (Team et al., 2024), on widely-used benchmarks (with commonly-used benchmark-specific prompt) MMLU (Hendrycks et al., 2021a) (5-shot), MMLU-Pro (Wang et al., 2024a) (5-shot, CoT), BBH (Suzgun et al., 2022) (3-shot, CoT), GSM8K (Cobbe et al., 2021) (4/8-shot, CoT), MATH (Hendrycks et al., 2021b) (0/4-shot, CoT), GPQA (Rein et al., 2024) (5-shot, CoT), HumanEval (Chen et al., 2021) (0-shot), MBPP (Austin et al., 2021) (0-shot). The performance is provided in Table 5. First, LMNet-1B brings comprehensive and significant improvement over the vertex model Qwen2.5-0.5B. This gives a way to improve general performance given a pre-trained LLM, using public data and acceptable training cost. Second, comparing LMNet-1B with other open-source pre-trained LLMs with similar or slightly larger size, LMNet-1B shows comparable or even better performance. This gives an efficient and effective way to scale for general intelligence by utilizing existing pre-trained LLMs rather than train single LLM from scratch.

Note that we aim at providing a new method rather than the trained LMNet-1B ready-to-use. As we had very limited computation budget, we expect further improvement, by larger-scale training with more deliberate data and schedule, deeper and wider LMNet structure, larger and more diverse vertex modules, and integrating post-training and reinforcement learning.

## B.2 Discussion about Scaling for General Intelligence

The pursuit of general intelligence through LLMs has traditionally focused on scaling individual models (Kaplan et al., 2020; Brown et al., 2020; Achiam et al., 2023; Grattafiori et al., 2024; Yang et al., 2024; Team et al., 2025), either by increasing their parameter count or enhancing their training data. However, this approach faces diminishing returns due to computational costs of the challenges of further optimizing monolithic architectures, and the gradual consumption of high-quality data. Our proposed LMNet paradigm offers a alternative by leveraging existing pre-trained LLMs as modular components within a densely connected network.

From a technical perspective, comparing with training a single LLM, LMNet has advantage in training cost and data requirement. Note that people need to train from scratch to build a larger LLM, and keep requiring new high-quality data to enhance performance. The first problem is training from scratch is wasting existing pre-trained LLMs which have already encoded vast information. LMNet addresses this by utilizing pre-trained LLMs in vertexes. The second problem is high-quality data is gradually depleted. LMNet addressed this by that LMNet can be trained with the data that has been used for training vertex LLMs, to enable the communication path and adapt transformer to dense vector inputs. Note that existing collection of LLMs could not be optimized for general intelligence effectively, because they communicate through natural language, disabling large-scale efficient optimization through gradient-descent.

From a motivational perspective, by taking pre-trained LLMs as relative lower-level intelligence, LMNet embodies the collective intelligence, the important idea that has been widely claimed and practiced. In LMNet, each vertex (a stripped transformer from pre-trained LLM) acts as a modular component, while the edges (trainable seq2seq modules) facilitate differentiable communication, enabling the system to collectively solve problems beyond the reach of any single LLM. By optimizing these communications, it can be expected that LMNet not only scales computational power but also fosters emergent behaviors, such as advanced reasoning and adaptive problem-solving, that transcend the capabilities of its individual components. Note that with more computation budget, one can also increase the intelligence versatility by implementing different vertexes without parameter-sharing, with LLMs from different expertise/families/sizes.

## C Details of Analysis of the Learned Dense Communication

### C.1 Visualization the Learned Edges

We try to find some topology patterns across the edges. We pick out the parameters in all edge modules of the edges connecting the 1/4/4/4/1 vertexes, for visualization and analysis (query/key/value/output projection matrix respectively). As provided in Figure 456 and 7, they show similar statistical

patterns: though they are different and from each other and initialization, it is hard to distinguish which edge is more significant. We can draw the following conclusions:

• All edges are obviously different from each other, and statistically different from the random initialization.

• No significant pattern among the edges, of statistical information (including std, max/min singular value) of each edge's parameters, can be witnessed.

• Above results indicate that the layer-wise fully connected structure is fully exploited through training, that it does not collapse to a simpler structure, e.g., chains or trees.

Note that above results do not indicate the layer-wise fully connected structure is necessary or optimal. A LMNet with other topology hypothesis could show similar results.

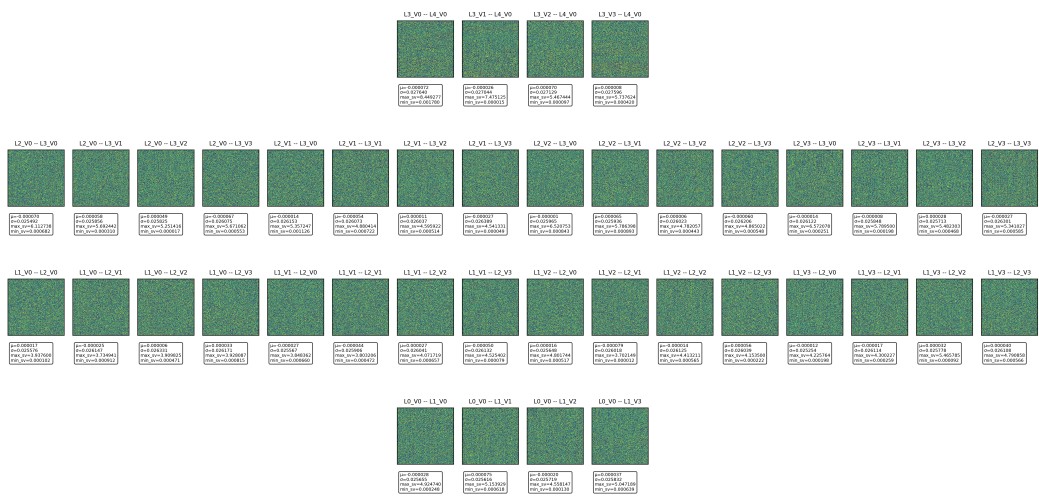

Figure 4: Visualization of query projection matrix of the attention block on every edge in trained LMNet. All edges are shown under the same value-color mapping.

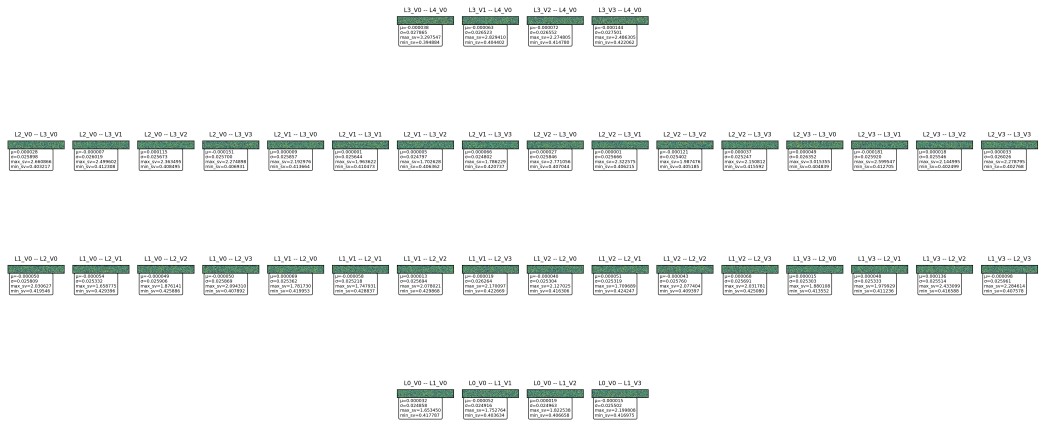

Figure 5: Visualization of key projection matrix of the attention block on every edge in trained LMNet. All edges are shown under the same value-color mapping.

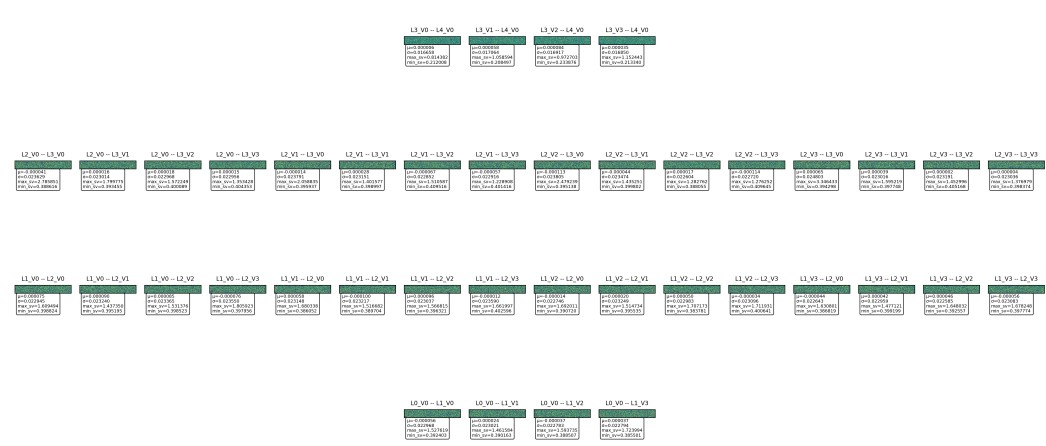

Figure 6: Visualization of value projection matrix of the attention block on every edge in trained LMNet. All edges are shown under the same value-color mapping.

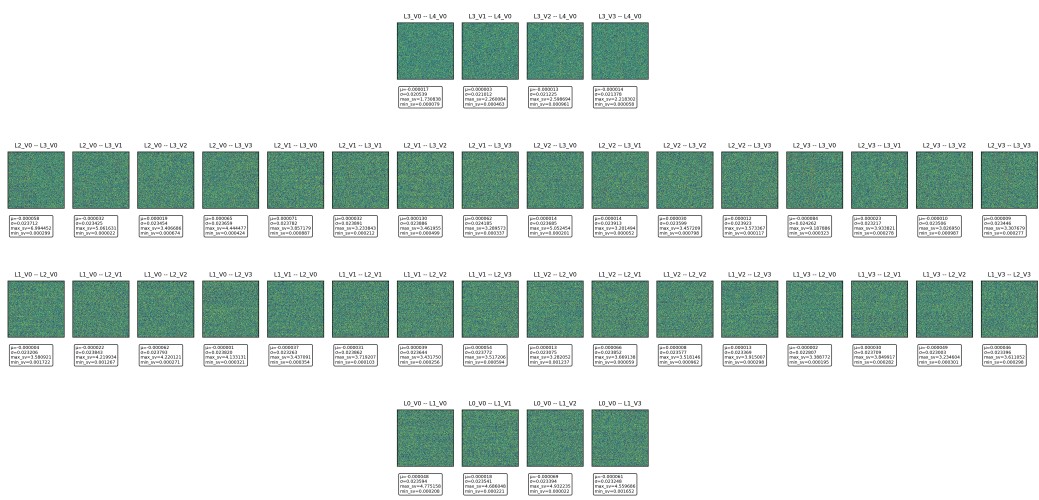

Figure 7: Visualization of output projection matrix of the attention block on every edge in trained LMNet. All edges are shown under the same value-color mapping.

## C.2 CASE STUDIES

The output vectors of intermediate vertexes, i.e., the input vectors of edge modules, can be de-embedded to meaningful sentences. To obtain the output of certain intermediate vertex, we feed the the intermediate output back to the input vertex auto-regressively, i.e., cutting off all vertexes at latter layers and the other vertexes at the same layer, while keeping all former modules. We de-embed the output sentences of all the 1/4/4/4/1 vertexes and process visualize: truncation begins at the position at input length and ends at first <eos> token; escape characters are removed for readability.

Table 6: Performance comparison of varying training data tested on GSM8K dataset (Accuracy, %).

|  | Qwen2.5-0.5B | Qwen2.5-1.5B |
|---|---|---|
| Prompt | 41.6 | 68.5 |
| SFT (7.47k training) | 24.1 | 51.1 |
| LMNet (7.47k training) | 30.9 | 60.0 |
| SFT (7.47k+93.7k training) | 42.9 | 69.3 |
| LMNet (7.47k+93.7k training) | 50.8 | 72.6 |

Table 7: Performance comparison on GSM8K dataset (Accuracy, %).

|  | Llama3.2-1B | Llama3.2-1B-Instruct | Qwen2.5-0.5B | Qwen2.5-1.5B |
|---|---|---|---|---|
| Pred | 2.88 | 30.10 | 5.31 | 9.25 |
| Prompt | 11.49 | 43.67 | **41.60** | **68.50** |
| SFT | 25.32 | 35.63 | 24.11 | 51.08 |
| LMNet | **33.41** | **45.75** | 30.93 | 60.02 |

# D CUSTOMIZING WITH VARYING DATA, AND ON CHALLENGING BENCHMARKS

The training set of GSM8K is very small (7.47k sentence). If trained on such a small training set only, it will seriously constrains the effect of methods that updating model parameters (including LMNet and SFT). We provide the performance under such circumstance in Table 7. Given Llama3.2-1B or Llama3.2-1B-Instruction as backbone LLM, LMNet still performs the best. However, given Qwen2.5-0.5B or Qwen2.5-1.5b, LMNet fails in comparison with Prompt, and the performance of Prompt and LMNet on Llama3.2-1B-Instruction are close. We infer the following two factors together cause such result. First, the training set of GSM8K is much smaller than MMLU (7.47k sentences vs 100k sentences). This result in overfitting risk for methods that updates model parameter, including SFT and LMNet. Despite of this factor, comparing with SFT, LMNet shows advantage, summarized as Table 6. So we have report add additional training data in the experiments reported in main text (Table 3).

Second, these two LLMs are strong enough for such tasks with a single model given proper instructions, that the additional communication steps brought by LMNet do little help. This can also be witnessed in Table 2, from the relative close performance between different methods on these three LLMs comparing with the other weak LLMs. So we further make verification that LMNet would show more advantage on more challenging benchmarks. We test the performance on OpenR1-Math-220k (an unseen subset, much more challenging than GSM8K). Table 8 shows the results. We find that the performance gain of LMNet is much significant comparing with easy tasks and comparing with other baselines, which indicates that LMNet would have larger effect on more challenging tasks, with more necessity of communication.

Table 8: Performance comparison tested on OpenR1-Math-220k dataset (Accuracy, %).

|  | Qwen2.5-0.5B | Qwen2.5-1.5B |
|---|---|---|
| Prompt | 18.0 | 29.0 |
| SFT (7.47k+93.7k training) | 23.2 | 34.7 |
| LMNet (7.47k+93.7k training) | 29.0 | 46.0 |

# E  INTEGRATING LMNET WITH PEFT METHODS

Table 9: Performance comparison on E2E dataset with GPT2-M. * indicates results from (Hu et al., 2022).

| Method | Metrics ↑ | | | | | Rank Avg. |
|---|---|---|---|---|---|---|
| | BLEU | NIST | MET | ROUGE-L | CIDEr | |
| No Adaptation | 0.00 | 0.42 | 0.04 | 0.16 | 0.00 | 12.0 |
| FT* | 68.2 | 8.62 | 46.2 | 71.0 | 2.47 | 5.6 |
| Adapter$^L$(0.37M)* (Lin et al., 2020) | 66.3 | 8.41 | 45.0 | 69.8 | 2.40 | 9.6 |
| Adapter$^L$(11.09M)* (Lin et al., 2020) | 68.9 | 8.71 | 46.1 | 71.3 | 2.47 | 5.2 |
| Adapter$^H$* (Houlsby et al., 2019) | 67.3 | 8.50 | 46.0 | 70.7 | 2.44 | 8.2 |
| FT$^{Top2}$* (Li & Liang, 2021) | 68.1 | 8.59 | 46.0 | 70.8 | 2.41 | 7.8 |
| PreLayer* (Li & Liang, 2021) | 69.7 | 8.81 | 46.1 | 71.4 | 2.49 | 4.2 |
| LoRA (Hu et al., 2022) | 68.9 | 8.68 | **46.5** | 71.5 | 2.51 | 3.4 |
| LMNet | **70.5** | 8.85 | **46.5** | 71.5 | 2.48 | **2.2** |
| LMNet + FT | 66.0 | 8.46 | 42.4 | 68.3 | 2.05 | 10.2 |
| LMNet + Prefix | **70.5** | **8.88** | 46.2 | **72.4** | 2.46 | 2.4 |
| LMNet + LoRA | 70.1 | 8.82 | 46.2 | 71.7 | **2.54** | 2.4 |

To avoid over-fitting with LMNet, by default we only train the edge parameters and keep the vertex parameters frozen. We still construct LMNet with a small scale (3 layers with 1/2/1 vertexes), using an attention block, containing 5.25M parameters, for each edge module. Note that thanks to the fully-differentiable paths in LMNet, we can also integrate LMNet along with PEFT methods. For example, we implemented LMNet+Prefix (Li & Liang, 2021) by adding random-initialized and to be adapted prefix before the initial input $\mathbf{X}^{in}$. LMNet+FT and LMNet+LoRA mean not only updating edge parameters, but also updating vertex parameters in a fully-adaptation or low-rank adaptation manner respectively. The results are provided in Table 9. In terms of the average value of rankings under every evaluation metric, LMNet performs best, showing learning dense communication makes effective and generalizable adaptation. LMNet+Prefix wins under the most individual metrics, suggesting plugging-in appropriate PEFT methods can further improve the performance.

