# OpenReview forum: "Learning Communication between Language Models through Dense Vectors"
_ICLR.cc/2026/Conference — Submitted to ICLR 2026_

### Official Review · Reviewer_2ZRn · 2025-10-29

**Soundness:** 2
**Presentation:** 3
**Contribution:** 3
**Rating:** 4
**Confidence:** 4

**Summary:**

This paper proposes LMNet, a graph-style architecture where multiple pre-trained LLMs are stripped of their embedding/de-embedding layers to form vertex transformers, which communicate via small trainable edge seq2seq modules carrying dense vector messages. The authors instantiate LMNet with shared vertex weights and end-to-end autoregressive training, then evaluate two settings: (i) “general intelligence” improvements using a 1.1B-parameter LMNet built from Qwen2.5-0.5B, trained on public data, and (ii) data-limited customization where edges are trained and compared to PEFT baselines like LoRA on MMLU, GSM8K, and E2E. Results show sizable gains over Prompt/SFT and competitive performance versus similarly sized monolithic LLMs.

**Strengths:**

- Recasts inter-model communication as learned dense messaging rather than natural-language tokens, enabling end-to-end optimization across models and edges; the layer-wise fully connected topology + edge translators is interesting. The overall idea is conceptually novel.
- Method is well specified: vertex/edge definitions, aggregation by sum, and a training recipe that first optimizes edges, then all parameters.
- Evaluated on diverse benchmarks.
- Provided case studies for de-embed intermediate states to probe what’s carried on the “wires”.

**Weaknesses:**

- One set of experiments I believe is missing is that the performance comparison between different width and depth of the LMNet, the results will be more convincing if there is plot showing Num of vertexes v.s. Performance, and showing that the performance positively scales with the vertex network size.
- One stated motivation (replace inefficient NL messages in multi-agent systems) doesn’t really match the implemented setup (single final decoder; interior modules pass only the prompt sequence). What they’ve actually built/benchmarked is much closer to stacked, cross-connected transformer blocks that exchange dense features before any token is produced, not agents sending complete messages to one another.

**Questions:**

- Typo: line 172 missing a space between ‘single’ and ‘X’.
- In Line 217-219: the author mention an alternative where each vertex could auto regressively generate multiple token embedding sequence, by didn’t specified how. In the normal LLMs, such decoding is controlled by the EOS token so the LLM knows when to stop the autoregressive process, I am curious how to do that without the decoding of EOS in the intermediate layers?
- A fair comparison shouldn’t only be about training compute; it should also hold test-time compute fixed. LMNet’s per-token inference effectively runs the vertex transformer N (i.e., number of vertexes in the net) times, so a base model given an equal test-time budget could use test-time scaling tricks to spend similar compute and might close some of the gap. The paper acknowledges LMNet increases inference latency per token roughly with layer depth L (sequential) even if same-layer vertices parallelize, but it doesn’t benchmark compute-matched inference baselines.

---

> ### Author Response · Authors · 2025-11-20
>
> Thanks a lot for your valuable comments.
> We appreciate your careful reading a lot, and find your review especially professional and constructive.
> We respectfully request you to first read the General Response first. Apart from the General Response, we want to reply the following point:
>
> ## Discussion about how to implement the alternative where each vertex could auto regressively generate multiple token embedding sequence.
> As you have mentioned, currently the end-to-end auto-regressive implementation indeed does not exactly match the image of replacing inefficient NL messages in multi-agent system. The current performance gain should be attributed to more test-time computation, where each vertex has been equipped with some newly assigned function through training, rather than the imaged sentence-by-sentence communication manner. This is consistent with our observation in Sec 4.1.3, that no semantically readable sentence can be found if de-embed intermediate outputs following the end-to-end auto-regressive manner. The readable sentence in Fig 3 can only be obtained by process mentioned in line 913~917, which does not exactly matches the inference process of LMNet, but it indicates that the NLP function of vertex modules are not completely destructed, and through connected following different edges, different vertexes can perform different functional roles.
>
> To exactly match the image of replacing inefficient NL messages in multi-agent system, i.e., sentence-by-sentence communication, one way is as we suggested in line 217~219.
>
> If keep 'sum' as aggregation, for implementation in practice, we suggest that the output 'sentence' of all vertexes except the last one, are forced to fill output_length $M$. While the layer-wise max_length setting $M_l$ should be set monotonically increasing with layer index $l$. For example, we can set max input length $M_0=512$, the output length of first layer $M_1=1024$, $M_2=2048$... Consider a vertex at first layer, with a <512 length input. It generate 'sentence' of dense vectors with length $M_1$, where the latter tokens might means 'EOS' or nothing, but during training we do not supervise it, leaving it to be processed by next module. Such implementation makes the 'sentence' output by the same layer aggregatable by sum. It might be also possible that only feed the last $M_l-M_{l-1}$ tokens generated by $l$-th layer to the $l+1$ layer to avoid length accumulation for efficiency, as if the old information is important, we have many vertexes in one layer where some vertex should learn to copy it. In such case, a $\Delta M_l$ should be set rather than $M_l$, with out the requirement to be monotonically increasing.
> Only the final output vertex at last layer, are supervised by text and using EOS token like ordinary.
>
> Another consideration we have had is that we might not use 'sum' as aggregation, because it can result in information loss. We can use 'concatenate' for aggregation, like reading language. As the topology of LMNet architecture is pre-defined, this is also implementable by setting the order among vertexes in the same layer, and scheduling output_length in different layers.

---

### Official Review · Reviewer_KV29 · 2025-10-30

**Soundness:** 3
**Presentation:** 3
**Contribution:** 2
**Rating:** 4
**Confidence:** 3

**Summary:**

The paper presents LMNet, a new method for learning communication between LLMs besides exchanging natural language token. The key idea is to strip out the embedding and de-embedding layer and treat those as vertices, and connect them with trainable seq2seq edge modules, forming a fully connected directed graph that is differentiable e2e.

The authors demonstrates two applications, showing increase in both general intelligence/reasoning task (MMLU, GSM8K etc.) and they also showed that LMNet can be used to train with customized, small scale data.

**Strengths:**

1. Reimagining LLM communication: the paper makes a very clean observation that current multi-LLM systems still talk in discrete natural language, even though the models internally think in dense vectors. That forces every intermediate module to do an unnecessary (and non-differentiable) de-embed → embed step, which is bad both for information efficiency and for gradient flow. The proposed fix is easy to understand. It also clearly distinguishes this work from latent CoT papers, which stay inside a single model’s latent space, whereas this paper explicitly targets inter-model communication.

2. Nice empirical signal: The cost is well-justified and the performance gain on benchmarks like MMLU and GSM8K. This justify for the better information flow that LMNet is designed for.

**Weaknesses:**

1. Limited ablation: The design of stacking a fully-connected seq2seq module is not studied in depth and seems a bit arbitrary. It remain unclear what the source of gain is. The authors should conduct ablation studies on different topology and putting different capacity of edge modules. Could you show that the design of the specific components of the architecture is actually useful? Like replacing seq2seq module with pure MLP/using a sparse topology.

2. Parameter size as confound and scalability issue: In the main experiment the author points out that the LMNet variant of Qwen-0.5B actually become 1.1B and so this intuitively feels like the performance gain is almost guaranteed. Although the authors compare to models with similar size, it isn't an apple-to-apple comparison with non-Qwen models as for many benchmarks the base Qwen-2.5-0.5B is already better, and versus Qwen2.5-1.5B it's still quite lagging on MMLU and GSM8K. Also, maybe I'm understanding this in a wrong way but does that mean for larger models (say 70B), the LMNet variant would become even bigger?

3. Communication not studied: The main experiment used the same model on the vertices but communication between LLMs are usually with different sharer and receiver. I'm wondering if this architecture generalizes to different models on vertices.

**Questions:**

See weaknesses.

---

> ### Author Response · Authors · 2025-11-20
>
> Thanks a lot for your valuable comments. We respectfully request you to refer to the General Response.

---

### Official Review · Reviewer_VPwU · 2025-10-31

**Soundness:** 2
**Presentation:** 2
**Contribution:** 2
**Rating:** 4
**Confidence:** 2

**Summary:**

The paper proposes LMNet, a method for connecting multiple language models through continuous dense vectors instead of discrete tokens, treating LLMs as vertices in a differentiable graph with trainable edge modules for communication. The key innovation is removing embedding/de-embedding layers between consecutive LLMs, allowing hidden states to flow directly between models through trainable "edge" modules. The authors construct a directed graph architecture where vertices are stripped transformers (without embedding layers) and edges are small seq2seq modules (typically single attention blocks). To reduce parameters, they employ parameter sharing where all vertices use the same pre-trained LLM.

The paper demonstrates two applications: (1) improving general intelligence by training a 1.1B parameter LMNet based on Qwen2.5-0.5B vertices, achieving ~40% relative performance gains with <0.2% additional training cost as claimed in the paper, and (2) data-efficient domain adaptation, where LMNet outperforms fine-tuning and latent reasoning methods on MMLU, GSM8K, and E2E benchmarks.

**Strengths:**

- The research question in this paper is interesting. The framing of computational graphs is novel and enables end-to-end gradient-based optimization.
- The paper demonstrates improvments on general tasks, math reasoning, knoweldge benchmarks and domain adaptation.

**Weaknesses:**

- LMNet is compared against its base model, making it unsurprising to gain performance improvement. Would be better if a model with comparable size is compared. Llama3.1-1B is provide in the paper, but its performance is even worse than the base model which should be problematic.
- The 5-layer architecture also introduce signifcant lower inference compared to a single pass.
- Performance on some benchmarks such as ARC-C, GPQA shows minimal gains or even degradation compared to Qwen2.5-1.5B.
- Passing hidden states between models is not new considering latent reasoning. But it is a bit overstated by claiming as "beyond human constraints". It's important to provide concrete evidences that dense vector communication can beat prompt-based approaches.
- The performance on communications between two different models are unknown especially these with different hidden dimensions.
- What happens at larger models such as 7B models? Does parameter sharing still work?

**Questions:**

Please see weakness

---

> ### Author Response · Authors · 2025-11-20
>
> Thanks a lot for your valuable comments. We respectfully request you to refer to the General Response.

---

### Official Review · Reviewer_yHkZ · 2025-11-01

**Soundness:** 2
**Presentation:** 2
**Contribution:** 1
**Rating:** 2
**Confidence:** 4

**Summary:**

This paper proposes a new paradigm for communication between language models by directly exchanging dense continuous vectors instead of natural-language tokens. The authors construct a directed graph, LMNet, where each vertex is a language model and each edge is a trainable seq2seq mapping that learns how to translate hidden representations between models. The entire structure is optimized end-to-end via gradient descent. The paper argues that this dense communication removes redundant embedding/de-embedding steps and enables differentiable multi-model cooperation. Two illustrative applications are presented: (1) enhancing general reasoning ability and (2) customizing models with limited data.

**Strengths:**

The idea of representing inter-model communication as differentiable dense vectors is conceptually novel and could inspire new research on multi-agent LLM systems.

The proposed LMNet graph abstraction provides a potentially unifying framework for studying information flow between models.

The paper touches on an interesting question of whether model interactions must occur through natural language at all, which is intellectually provocative.

**Weaknesses:**

The motivation for removing the token layer is unconvincing. Tokenization does not necessarily cause semantic loss, while replacing discrete tokens with dense vectors may introduce additional noise, instability, and loss of interpretability.

No clear empirical evidence demonstrates that dense communication improves performance, efficiency, or convergence compared with existing approaches (e.g., natural-language interaction, hidden-state distillation, or adapter-based transfer).

Experiments are limited to small-scale toy settings without strong baselines or ablation studies, making it difficult to assess generality or practical benefit.

The paper lacks theoretical analysis of communication capacity, robustness, or scalability.

Overall, the work feels more like a conceptual proposal than a rigorously validated method.

**Questions:**

Can the authors provide quantitative comparisons showing improvements in reasoning accuracy, efficiency, or resource usage versus token-based communication?

How stable and interpretable are the learned dense communication vectors across tasks or model sizes?

Have the authors analyzed whether the introduced dense mapping modules amplify noise or reduce robustness?

---

> ### Author Response · Authors · 2025-11-20
>
> Thanks a lot for your valuable comments. We respectfully request you to first read the General Response. Apart from the General Response, we want to reply the following points:
>
> ## Lack of comparison with existing methods.
> For improving general intelligence, please refer to general response. For improving with specific training data, we have compared with latent-reasoning methods in Table 2,3.
> Please let us know if any specific additional baseline should be compared.
>
> ## Semantic loss comparing discrete tokens and dense vectors.
> Let's take Qwen2.5-0.5B as example for analyzing the semantic information carried by each token. With discrete vocabulary. each token carries information $log_2 (151936)\simeq17.21$ bits, where 151936 is the dictionary size. With dense vectors in float32, each token carries information $896\times 32=28672$ bits, where 896 is the embedding dimension and 32 is the bits of each dimension. Comparing 17 and 28672 bits, there is significant difference between the information each token can carry using discrete vocabulary/dense vectors.

---

### Author Response · Authors · 2025-11-20
**General Response**

We sincerely thank all reviewers for their valuable feedback.
We are glad to see that all reviewer are generally consistent with acknowledging the novelty of the proposed method and conceptual contribution, and the main concerns about experiments.

## Experiment with larger scale.
__1.For the setting general intelligence (Sec 4.1):__

we fully agree, and have expected, a main concern is lacking experiments with larger scale, like using larger and diverse LLMs as vertex, or ablation the width and depth of LMNet. However, this is far beyond our capacity. Note that training existing LMNet-1.1B already requires about 80 GPUdays. And each single experiment above would require training a larger-scale LMNet from scratch(edge). This is only feasible with industrial scale resources. This will take years for us which is impossible to address through rebuttal or any later manuscript in our academic capacity. We hope the reviewers could understand such fact. It would be a pity that this paper could not be able to shown to wider audience through this conference due to such reason.

__2.For he setting customizing LLM with limited data (Sec 4.2):__

we have had experiments on various models in Table 2 and 3. We add the following results using 3B and 7B models follwomg the same setting in 4.2.1, which shows consistent advantage:
MMLU ($\Delta$ ACC %):|  Qwen2.5-3B| Qwen2.5-7B|GSM8K (ACC %):|  Qwen2.5-3B| Qwen2.5-7B|
-|-|-|-|-|-
Prompt|43.6|52.1|\|\||84.2|90.4
SFT(w/ CoT)|43.8|49.5|\|\||86.0|90.5
LMNet|__45.2__|__53.8__|\|\||__87.7__|__92.3__


For ablation on depth/width of LMNet architecture, due to computation budget, we turn to using GPT2-M on E2E dataset (the same as 4.2.2), no significant difference or patterns can be observed in this range:
Architecture|  # Training Parameters (edge)| BLEU|NIST|MET|ROUGR-L|CIDEr
-|-|-|-|-|-|-
LMNet-1/1|21.00M|69.1|8.76|__46.6__|70.6|2.43
LMNet-1/2/1(reported)|36.75M|__70.5__|__8.85__|46.5|71.5|2.48
LMNet-1/2/2/1|57.75M|69.8|8.80|46.0|71.1|__2.49__
LMNet-1/4/1|57.75M|69.1|8.70|45.9|__71.6__|2.44
LMNet-1/4/4/1|141.75M|68.7|8.79|46.5|71.2|2.48

## Fair comparison with base model.
More than one reviewers have mentioned that in our experiment in Table 1, comparing the left part: 0.5B actually become 1.1B and so this intuitively feels like the performance gain is almost guaranteed; while comparing the right part: LMNet-1.1B has no obvious advantage.
We want to emphasize that the additional (1.1-0.5)B parameters in LMNet is introduced with is far less cost than the similar scale parameters inside LLM.
As reported in lin 288 and Appendix B, while training a LLM from scratch takes a scale at 10^1 T tokens training, the we only takes a scale at 10^(-2) T tokens to train LMNet to reach such performance.
So we want to show that LMNet can significantly improve vertex model's general performance by training additional parameters, to comparable with similar scale(parameter-in-total) monolithic LLM, while the total training cost is far less.

A very rational comparison we want to introduce here is comparing LMNet with TTS using the same base model, as they both increases the inference-time computation. We add experiments comparing LMNet-1.1B with representative parallel TTS strategy Self-Consistency and sequential TTS strategy Self-Refine. As LMNet-1.1B contains 1+4+4+4+1=14 repeats of Qwen2.5-0.5B and additional 0.7B edge modules (no repeat), we compare with above TTS with a budget of 16 trials/iterations using Qwen2.5-0.5B.
The following table shows the results, where LMNet significantly outperform the other. Of course, LMNet requires additional training comparing with the others, but the cost is near zero comparing with the cost of training a monolithic LLM. Note that while there are also existing TTS with additional training using RL or training an additional verifier, they are specific to certain benchmark using corresponding curated training dataset. In fact, one of the motivations of LMNet is exactly it can learn to express TTS-like process through the MLP-like  network structure, through training the higher-level network with general data.

Method/Benchmark |  MMLU| MMLU-pro|BBH|GSM8K|MATH
-|-|-|-|-|-
Base(1)|44.3|15.7|20.3|41.6|19.5
Self-Consistency(16)|46.0|17.1|21.6|43.5|19.8
Self-Refine(16)|45.2|16.4|24.7|42.1|19.0
LMNet-1.1B(1/4/4/4/1)  | __53.9__ |__26.2__|__47.3__|__50.3__|__38.8__

---

### Meta-Review · Area_Chair_wB7L · 2026-01-09

**Summary:**

I have carefully checked the paper and review. The idea of dense and continuous communications between LLMs makes senses to me, but I agree with Reviewer yHkZ that it feels more like an interesting concept. Since many of the reviewer's concerns remain unaddressed, I have to reject this paper at its current form.

**Reviewer Concerns:**

The reviewers' concerns are mostly in (1) motivation of this dense inter-LLM communication; and (2) unconvincing empirical validation. I also agree with Reviewer VPwU and KV29 that the experimental results can be further strengthened.

**Reviewer Scores:**

All reviewers suggest rejection and no one changed the final score.

---

### Decision · Program_Chairs · 2026-01-26

Reject